# Elucidating the impact of Hypericum alpestre extract and L-NAME on the PI3K/Akt signaling pathway in A549 lung adenocarcinoma and MDA-MB-231 triple-negative breast cancer cells

**Hayarpi Javrushyan[1], Mikayel Ginovyan[1], Tigran Harutyunyan[2], Smbat Gevorgyan[3], Zaruhi Karabekian[4], Alina Maloyan[5], Nikolay Avtandilyan[1]***

**1** Research Institute of Biology, Yerevan State University, Yerevan, Armenia, **2** Department of Genetics and Cytology, Yerevan State University, Armenia, **3** Denovo Sciences Inc, Yerevan, Armenia, **4** Laboratory of Immunology and Tissue Engineering, L.A. Orbeli Institute of Physiology NAS RA, Yerevan, Armenia, **5** Center for Developmental Health, Knight Cardiovascular Institute, Oregon Health & Science University, Portland, Oregon, United States of America

* nv.avtandilyan@ysu.am

## Abstract

Plants of the Hypericaceae family have been traditionally used for their medicinal properties, including antibacterial, antiviral, and antioxidant activities. Among these, *Hypericum alpestre* (HA) extracts have shown notable cytotoxicity against various cancer cell lines, drawing attention to their phenolic compounds as potential anticancer agents. Similarly, N(G)-Nitro-L-arginine methyl ester (L-NAME), an inhibitor of nitric oxide synthase (NOS) activity, has emerged as a promising candidate in cancer therapy. However, the precise molecular mechanisms underlying the anticancer effects of both HA and L-NAME remain unclear. This study aimed to clarify the impact of HA and L-NAME on the phosphoinositide 3-kinase (PI3K)/protein kinase B (Akt)/ mammalian target of rapamycin (mTOR) signaling pathway in A549 human lung adenocarcinoma and MDA-MB-231 breast cancer triple-negative cells, with particular emphasis on the tumor necrosis factor-alpha (TNFα)/ cyclooxygenase-2 (COX-2) and vascular endothelial growth factor A (VEGFα)/matrix metalloproteinase-2 (MMP-2) pathways. *In silico* analyses identified compounds within HA extracts with the highest affinity for PI3K/Akt, a finding subsequently confirmed by *in vitro* experiments. Notably, the combination of HA and L-NAME demonstrated greater efficacy than the combination of HA and 5-fluorouracil (5-FU), as evidenced by enhanced apoptotic activity. Both HA alone and in combination with L-NAME inhibited the TNFα/COX-2 and VEGFα/MMP-2 pathways. These results suggest that the therapeutic effects of HA, especially in combination with L-NAME, may be mediated through the PI3K/Akt signaling pathway. A better understanding of the interaction between HA polyphenols and PI3K/ Akt signaling could pave the way for novel therapeutic strategies against cancer, including drug-resistant tumors.

**Data availability statement:** The data used to support the findings of this study are included in the articles. The minimal anonymized dataset necessary to replicate our study findings is available in Figshare. The dataset can be accessed using the following DOI: https://doi.org/10.6084/m9.figshare.28232882.v1https://doi.org/10.6084/m9.figshare.28234607.v1https://doi.org/10.6084/m9.figshare.28234586.v1https://doi.org/10.6084/m9.figshare.28234601.v1https://doi.org/10.6084/m9.figshare.28234631.v1" Along with the following Supporting Information file 'S1 File.rar'.

**Funding:** 1. Nikolay Avtandilyan: Recipient 23LCG-1F010: Grant number Armenia: Country The RA Ministry of Education, Science, Culture and Sports Higher Education and Science Committee: Organization 2. Tigran Harutyunyan 21AG-1F068: Grant number Armenia: Country The RA Ministry of Education, Science, Culture and Sports Higher Education and Science Committee: Organization 3. Hayarpi Javrushyan 24WS-1F036: Grant number Armenia: Country The RA Ministry of Education, Science, Culture and Sports Higher Education and Science Committee: Organization.

**Competing interests:** The authors have declared that no competing interests exist.

# 1. Introduction

The Hypericaceae family's plants are widely employed for various medicinal purposes, including the treatment of wounds, bruises, dysentery, acute mastitis, jaundice, hepatitis, sore furuncles, skin inflammation, nerve pain, hemoptysis, epistaxis, metrorrhagia, irregular menstruation, burns, hemorrhages, and a range of tumors. In addition, different extracts from these plants have been reported to possess antibacterial, antiviral, antibiotic-modulating, and both antioxidant and prooxidant properties [1–5]. In our earlier studies, when screening ethanolic extracts from eleven wild plant species for cytotoxic properties, *H. alpestre* (HA) extract emerged as one of the most potent inhibitors of A549 (lung adenocarcinoma) and HeLa (cervical carcinoma) cell growth [5]. It is well known, that phenolic compounds, as the most prevalent plant secondary metabolites, have attracted considerable attention due to their antioxidant/prooxidant properties and their potential role in the mitigation of diverse diseases associated with oxidative stress, including cancer. A total of 244 constituents were identified in *H. alpestre* extracts in our earlier research [6]. The metabolomic analysis of HA aerial part extracts using the UHPLC-ORBITRAP-HRMS technique unveiled the presence of several major phenolic constituents, suggesting that these substances might contribute to its high cytotoxic effects [6]. Considering the strong *in vitro* growth inhibiting properties of HA extract against different cancerous cell lines including the breast cancer model [5,6]. In the fight against cancer, there is a constant effort to discover new compounds that can have specific and targeted effects and have as few side effects as possible. Currently, there is significant interest in the development of combined cancer treatments using natural compounds with chemotherapeutic agents for their modulating, drug resistance modifying properties, or reduction of chemotherapy side effects [7]. Phytochemicals can affect cellular processes and signaling pathways, which have potential antitumor properties [7].

N(G)-Nitro-L-arginine methyl ester (L-NAME) is an inhibitor of nitric oxide synthase (NOS), a key enzyme in several biological pathways [8] . It belongs to a group of compounds that modulate enzyme activity. L-arginine is the primary substrate for enzymes like arginase and NOS, which are linked to cancer progression [9]. This relationship underscores the potential impact of substances like L-NAME on cancer-related biological processes, highlighting the complex interplay between amino acids and enzyme function in the context of cancer [10]. Increased NOS expression is associated with different cancers such as cervical, breast, lung, brain, and spinal cord cancers [11]. Inhibition of NOS activity has been suggested as a potential tool to prevent breast cancer [12] . In our previous research, we showed that *in vivo* inhibition of NOS activity by L-NAME influences the L-arginine pathway, specifically nitric oxide and polyamine biosynthesis, which have an inhibiting effect on cancer progression [6,11,13]. For the first time, we performed a combined treatment of breast cancer with HA herb extract and L-NAME in an *in vivo* experimental model of rat breast cancer [6]. This combination decreased tumor multiplicity, increased the amount of interleukin-2 (IL-2) in the tumor environment, activated antioxidant enzymes, and decreased histological score and tumor blood vessel area. The molecular mechanisms and key cellular players, whose modification manifests such an anticancer effect, have not been clarified. In this work, we addressed the effects of *H. alpestre* and L-NAME on the PI3K/Akt/mTOR-mediated changes in A549 lung adenocarcinoma and MDA-MB-231 triple-negative breast cancer cell cultures by elucidating the roles of the tumor necrosis factor-alpha (TNFα)/cyclooxygenase-2 (COX-2) and vascular endothelial growth factor A (VEGFα)/matrix metalloproteinase-2 (MMP-2) tandems.

The PI3K/Akt/mTOR signaling pathway is a major signaling pathway in various types of cancer [14,15]. It controls the hallmarks of cancer, including cell survival, metastasis, and

metabolism. The PI3K/Akt pathway also plays essential roles in the tumor environment, functioning in angiogenesis and inflammatory factor recruitment. The study of PI3K/Akt networks has led to the discovery of inhibitors for one or more nodes in the network, and the discovery of effective inhibitors is important for improving the survival of patients with cancer. To date, many inhibitors of the PI3K/Akt signaling pathways have been developed, some of which have been approved for the treatment of patients with cancer in the clinic [16,17]. However, many issues associated with the use of pathway inhibitors, including which drugs should be used to treat specific types of cancer and whether combination therapies will improve treatment outcomes, remain to be resolved. Activation of this pathway is related to many factors, including TNFa and VEGFa [18]. After activation by VEGF, Akt promotes the proliferation, migration, and survival of endothelial cells, thus affecting angiogenesis. A study showed that the PI3K/Akt signaling pathway promotes the development of inflammation by affecting neutrophils, lymphocytes, and other white blood cells [19]. IL-1, IL-6, TFN-α, and other inflammatory factors activate Akt and expand the range of inflammation, while Akt inhibition blocks both inflammation and tumor development [15]. Akt signaling promotes tumor cell survival, proliferation, growth, and metabolism by activating its downstream effectors such as cyclooxygenase 2 (COX-2), matrix metalloproteinases (MMPs), and Caspase-3. Multiple signaling pathways are involved in the regulation of tumor angiogenesis, among which PI3K/Akt signaling is the most important. PI3K forms a complex with E-cadherin, β-catenin, and VEGFR-2 and is involved in endothelial signaling mediated by VEGF through the activation of the PI3K/Akt pathway [20]. The PI3K/Akt signaling pathway also promotes TNF-induced endothelial cell migration and regulates tumor angiogenesis. MMPs and COX-2 also affect tumor angiogenesis [19,21]. In tumor invasion and metastasis, platelet-derived growth factor (PDGF) induces MMP expression through a PI3K-mediated signaling pathway. Upregulation of the antiapoptotic protein Bcl-2 and activation of the PI3K/Akt signaling pathway are the main mechanisms by which COX-2 stimulates endothelial angiogenesis. PI3K/Akt signaling blocks the expression of proapoptotic proteins, reduces tissue apoptosis, and increases the survival rate of cancer cells. Akt inhibits the proapoptotic factors Bad and procaspase-9 through phosphorylation and inhibits Caspas-3 activity [14,15].

Based on the aforementioned facts, we aimed to elucidate, through *in silico* methods, the polyphenols present in the HA extract that exhibit the highest affinity for the PI3K/Akt signaling pathway. To ensure the reliability of our data, we used *in vitro* A549 lung adenocarcinoma and MDA-MB-231 breast cancer triple-negative cell cultures to demonstrate the herb extract's effects on PI3K and Akt enzymes and clarify how PI3K/Akt/mTOR mediation regulates TNFα, VEGFα, COX-2, MMP-2, and Caspase-3. Since PI3K/Akt/mTOR signaling dysregulation plays a crucial role in cancer drug resistance, discovering new compounds targeting these components could help overcome drug resistance in various cancer therapies.

## 2. Materials and methods

### 2.1 Chemicals and reagents

All chemicals were purchased from Sigma-Aldrich (USA) and Abcam (UK). ELISA kits for TNFa (ab46087), VEGFa (ab193555), MMP-2 (ab92536), COX-2 (ab38898), PI3K and phosphorylated (p)-PI3K (ab191606), AKT and p-AKT (ab179463), P53 (ab46067) were purchased from Abcam. The following antibodies were used in the Western Blot analysis: Anti-Akt (EPR16798), anti-PI3K (EPR18702), anti-mTOR (EPR390(N)), anti-β-actin (ab8227) and Anti-caspase (EPR18297; all Abcam (UK)).

## 2.2 Plant material

The aerial parts of the *Hypericum alpestre* plant were harvested from the Tavush region of Armenia (at an altitude of 1600-2800 meters above mean sea level) during the flowering period, following the previously described protocol [22]. Identification of plant materials was carried out at the YSU Department of Botany and Mycology by Dr. Narine Zakaryan. The plant materials were deposited at the Herbarium of Yerevan State University (YSU), where they were assigned a voucher specimen serial number [6].

## 2.3 Plant crude extract preparation

The grounded plant materials were extracted by maceration with 96% ethanol at a 10:1 solvent-to-sample ratio (v/w). Stock solutions of 50 mg DW/mL crude ethanol extract were prepared as described earlier [5]. The percent yield was 10.60 ± 2.31%.

## 2.4 Cell cultures

Human lung adenocarcinoma A549 (cat # CCL-185) and triple-negative breast cancer MDA-MB-231 (cat # HTB-26) cell cultures were obtained from ATCC and maintained in DMEM medium supplemented with L-glutamine (2 mmol/L), sodium pyruvate (200 mg/L), fetal bovine serum (100 mL/L), and antibiotics (100 U/mL penicillin and 100 μg/L streptomycin). Cells were grown at 37 °C under a humidified atmosphere with 5% $CO_2$ in a Biosmart (Biosan, Latvia) as described before [23]. Cultured cells were regularly examined for the presence of mycoplasma contamination using the Universal Mycoplasma Detection Kit from ATCC (Manassas, Virginia, USA).

## 2.5 MTT cytotoxicity test

The inhibitory effects of the HA extract on the proliferation of A549 and MDA-MB-231 cell lines were assessed using the MTT assay, as previously detailed [5]. The assay was performed in three separate experiments, each containing four technical replicates. Cytotoxicity was determined by calculating the percentage of growth inhibition in cells treated with the plant extract or phytotherapeutic agent compared to control cells treated only with the solvent (resulting in a final concentration of 1% ethanol), which was designated as 100% growth. The IC50 values for the extract at different exposure times were calculated using a nonlinear regression analysis with a variable slope [24].

## 2.6 ELISA of TNFa, VEGFa, COX-2, MMP-2, p53, and Akt

A549 cells ($2 \times 10^5$ cells per well) were cultured in 12-well plates and incubated for 24 h. After incubation, the cell medium (630 μL) was replaced. The cells were treated with Phosphate-buffered Saline (PBS) and 1% Ethanol solution (Control, A549), 5-FU (40 μM), HA (0.25mg/mL), L-NAME (14mM, LN) and HA + L-NAME (0.25mg/mL + 14mM, HALN) for 24 h and then the culture medium was harvested. TNFa, VEGFa, and MMP-2 in the supernatant were quantified according to the manufacturer's instructions. Cells from each group were collected (trypsinized, neutralized, centrifuged), lysed on ice with Lysis buffer, collected in a centrifuge tube, and further lysed for 10 min. The supernatant was collected after centrifugation at 13,000 × g for 10 min at 4°C. Changes in the levels of COX-2 and Akt were measured using ELISA kits, according to the manufacturer's instructions. Protein concentration in cell culture medium and lysates were measured using the Bradford method. Each test sample (70 μL) was added from three biologically independent repetitions, involving three different cell passages, each containing two technical replicates.

## 2.7 Phospho-PI 3 kinase p85 + total In-Cell ELISA assay

A549 cells ($1.5\times10^4$ cells per well) were seeded in the 96-well plates treated for tissue culture. After 24 h incubation, the cell medium (180 μL) was refreshed. The cells were treated with 20 μL control or test compounds with the following final concentrations: PBS, 1% ethanol solution (Control, A549C), 5-FU (40 μM), HA (0.25mg/mL), L-NAME (14mM, LN) and HA + L-NAME (0.25mg/mL + 14mM, HALN). The calculations for seeding the cells were conducted to ensure approximately 80% confluency at the fixation time. After 24 h exposure, the medium was discarded and cells were fixed with 100 μL of 4% formaldehyde in PBS. Crystal Violet was used to stain cells for normalizing readings in 450nm for Phospho-PI 3 kinase p85 + Total. The measured OD450 readings were normalized for cell number by dividing each OD450 reading by the corresponding OD595 reading for the same well. This relative cell number was then used to normalize each reading. Total and phospho-PI 3 kinase p85 were each assayed in triplicate using the phospho- and total PI 3 Kinase p85 antibodies included in the PI 3 Kinase Kit. Phospho-PI 3 kinase p85 and Total PI3K levels were measured using an In-Cell ELISA kit (ab207484), according to the manufacturer's instructions.

## 2.8. Caspase-3/CPP32 colorimetric assay

A549 cells ($5 \times 10^5$ cells per well) were cultured in 6-well plates and incubated for 24 h. Then, the cell medium (900 μL) was refreshed and the cells were treated with 100 μL of PBS + 1% ethanol solution (control, A549C) or test compounds with the following final concentrations: 5-FU (40 μM), HA (0.25 mg/mL), L-NAME (14mM, LN), HA + L-NAME (0.25mg/mL + 14mM, HALN) and HA + 5- FU (0.25 mg/mL + 40 μM). After 24 h the cells were harvested. Each test sample (100 μL) was added from three biologically independent repetitions, involving three different cell passages, each containing two technical replicates. Cells were resuspended in 50 μL of chilled Cell Lysis Buffer and incubated on ice for 10 minutes. Then, cell lysate was centrifuged for 1 min (10,000 x g). After that supernatant (cytosolic extract) was transferred to a fresh tube and put on ice for immediate assay. Fold-increase in CPP32 activity has been determined by comparing these results with the level of the uninduced control. Optical density values were corrected taking into account the number of cells. All steps were performed according to the protocol presented in the Caspase-3/CPP32 Colorimetric Assay Kit (K106, BioVision) instructions.

## 2.9 Western blots

A549 and MDA-MB-231 cells ($5 \times 10^5$ cells per well) were cultured in 6-well plates and incubated for 24 h. Then, the cell medium (900 μL) was refreshed and the cells were treated with 100 μL of PBS + 1% ethanol solution (control) or test compounds with the following final concentrations: 5-FU (40 μM), HA (0.25 mg/mL), and HA + L-NAME (0.25mg/mL + 14mM, HALN). After 24 h the cells were harvested. Cells suspended in Rippa buffer supplemented with protease and phosphatase inhibitor cocktail (Abcam). Total protein concentrations in the fraction were determined using the bicinchoninic acid (BCA) (Bio-Rad Laboratories). Total proteins (20 μg) were separated on 4-20% precast linear gradient gels (Bio-Rad Laboratories), transferred to nitrocellulose membranes, and blocked with 5% (w/v) nonfat milk in TBST for 1 h. Membranes were incubated overnight at 4°C with the primary antibody diluted in 1% nonfat milk (w/v) in TBST and detected using an appropriate peroxidase-conjugated secondary antibody [25]. Products were visualized by ECL chemiluminescence (Millipore). Band intensities were measured using the Chemidoc (Bio-Rad Laboratories).

## 2.10 Morphological analysis of apoptosis by Hoechst 33258 staining

Hoechst 33258 staining enables the determination of the percentage of apoptotic cells through morphological analysis using a fluorescence microscope [26]. Briefly, A549 cells ($2 \times 10^5$ cells/mL) were incubated with vehicle or 5-FU (40 μM) as the positive control, and test samples: HA (0.25 mg/mL), LN (14 mM), HA + 5-FU (0.25 mg/mL + 40 μM), and HA+LN (0.25mg/mL + 14mM) for 24 h. Cells were fixed with 4% paraformaldehyde in PBS for 10 min, washed twice with PBS for 5 min, and stained with Hoechst 33258 (10 μg/mL) for 10 min in the dark. Then cells were analyzed under a fluorescence microscope (x250 magnification) (Zeiss, Germany). Cells with typical morphological nuclei changes, such as chromatin condensation, rough edges, nuclear fragmentation, and apoptotic bodies were counted as apoptotic. Controls and treatment variants were examined in duplicate. For each variant, 500 cells were scored and the percentage of apoptotic cells was calculated as follows: % apoptotic cells = (the number of apoptotic cells/500 cells)*100%.

## 2.11 Acquisition and analysis of protein structures

The structural elucidation of PI3K and AKT protein was conducted by retrieving their crystallographic forms from the Protein Data Bank (PDB), accessed at [https://www.rcsb.org/], with the specific PDB IDs being 6AUD and 2JDO, respectively. Advanced molecular visualization and analysis were performed using the PyMOL Molecular Graphics System (Schrödinger, LLC). This phase included preprocessing steps for removing non-essential entities such as solvent molecules, ions, and extraneous non-protein components. Concurrently, ligands associated with these protein structures were segregated, yielding pure protein structures for subsequent computational analysis. The refined protein models were earmarked for molecular docking studies, whereas the separated ligands were preserved for re-docking procedures in subsequent structural bioinformatics applications.

## 2.12 Enhanced molecular docking using AutoDock Vina

AutoDock Vina, an extensively utilized molecular docking tool, employs the Iterated Local Search global optimizer. This optimizer is conceptually akin to those used in ICM and the Broyden-Fletcher-Goldfarb-Shanno (BFGS) quasi-Newton algorithm for local optimization. The software integrates a hybrid scoring function that combines empirical and knowledge-based approaches to enhance docking precision and efficiency. For optimal docking performance, the 'exhaustiveness' parameter was set to 8, adhering to the standard protocols recommended by the software developers. Utilizing AutoDock Vina facilitated a comprehensive assessment of the binding affinities of key compounds extracted from *Hypericum alpestre* (S1 Table) with the target proteins, allowing for their prioritization based on predicted binding efficacy [27].

## 2.13 Statistic analysis

Results are presented as means ± SEM. We analyzed the data using either one-way ANOVA or its non-parametric counterpart, the Kruskal-Wallis test, depending on the results of the normality test. Subsequently, Dunn's test was employed to evaluate the statistical significance of the results for TNFα, VEGFα, MMP-2, COX-2, Caspase-3, and apoptosis rate. The significance of the results obtained for PI3K and Akt was assessed using two-way ANOVA and Tukey's multiple comparisons tests. Statistical analyses were performed using GraphPad Prism 8 software (San Diego, CA, USA), and a significance level of $p < 0.05$ was deemed statistically significant.

## 3. Results

### 3.1 The interaction of HA extract metabolites with PI3K and Akt

Our previous work revealed the presence of more than 200 active polyphenolic compounds in HA extract [6]. In this study, the interaction of these compounds and PI3k/Akt was clarified by *in silico* computation biology modulation methods. Two compounds exhibiting the highest affinity were identified. Our results detail the specific binding interactions of chrysoeriol glucuronide and pseudohypericin with PI3K and AKT proteins, as observed in our molecular docking simulations (Fig 1A–1D). For chrysoeriol glucuronide docked with PI3K, multiple hydrogen bonds were identified: the ligand formed a hydrogen bond with the backbone nitrogen of Ala805 at a distance of 3.14 Å, and additional hydrogen bonds with Ser806 and Thr886, indicative of a strong interaction potential (Fig 1B). The ligand also exhibited hydrophobic interactions with key residues including Trp812 and Met804, which are likely to contribute to the binding affinity and specificity. Docking chrysoeriol glucuronide with AKT resulted in a

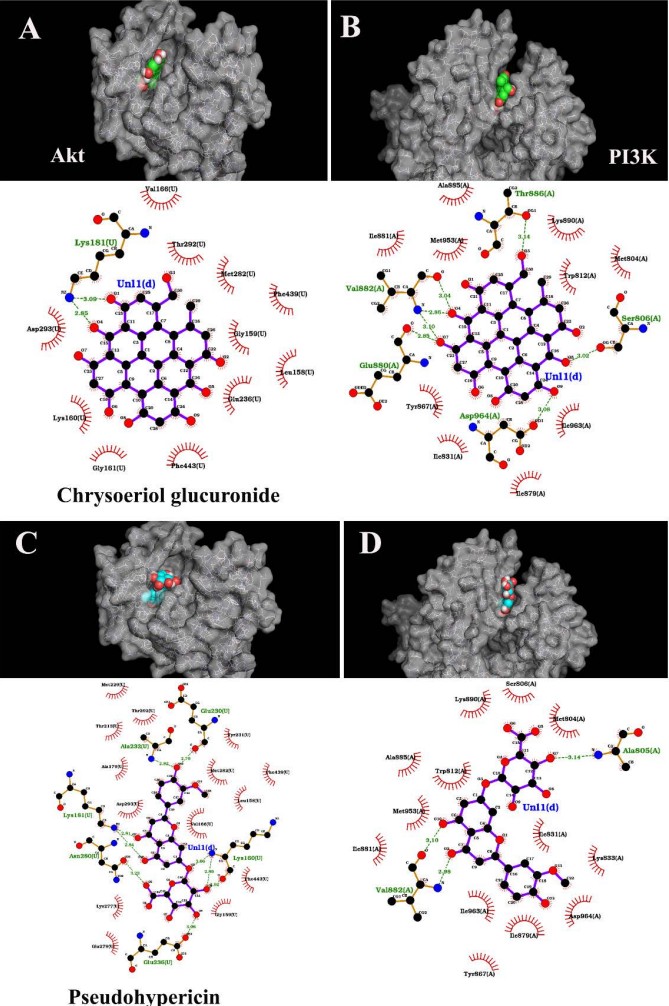

**Fig 1. 2D binding analysis and interaction types on PI3K (B, D) and AKT (A, C) in combination with 3D visualization.** A and B – Chrysoeriol glucuronide, C and D – Pseudohypericin. Hydrogen bonding is indicated by green dotted lines, while the remaining interactions are hydrophobic.

distinct interaction profile, with fewer hydrogen bonds observed (Fig 1A). Notably, a hydrogen bond with Glu230 at 2.70 Å was prominent, alongside hydrophobic contacts with Lys181 and Val166, suggesting an alternate binding conformation compared to its binding with PI3K. Pseudohypericin docking with PI3K revealed a different pattern: a network of hydrogen bonds with Glu880, Asp964, and Ile831, and hydrophobic interactions with residues such as Val882 and Tyr867 (Fig 1D). This pattern indicates a binding mode that is distinct from that of chrysoeriol glucuronide with PI3K. Finally, pseudohypericin docked with AKT showed an interaction pattern that includes hydrogen bonds with Asp293 and Lys160, and hydrophobic contacts with Glu236 and Met282 (Fig 1C). The interaction profile was again unique from its binding with PI3K and chrysoeriol glucuronide's interactions with both kinases.

These results highlight the specificity of ligand-kinase interactions and suggest that chrysoeriol glucuronide and pseudohypericin could potentially act as selective kinase inhibitors due to their ability to form multiple hydrogen bonds and hydrophobic interactions with distinct residues in PI3K and AKT (Tables 1 and 2).

## 3.2 Growth inhibiting properties of HA extract tested by MTT assay

The growth-inhibitory effects of HA extract at different concentrations and exposure times were studied on A549 and MDA-MB-231 cancer cells using the MTT assay. HA extract demonstrated a statistically significant ability to inhibit the growth of both tested cancer cell lines (Fig 2A and 2B). Shorter exposure times (4 hours) did not significantly inhibit the growth of MDA-MB-231 cells. However, a cytotoxic response was observed in A549 cells at HA extract concentrations starting from 0.25 mg DW/mL concentration. For A549 cells, exposure time had minimal effect on the growth-inhibiting properties of HA extract, whereas the effect increased in a concentration-dependent manner. In contrast, data obtained for MDA-MB-231 cells indicate that HA exhibits both concentration- and time-dependent cytotoxicity (Fig 2A and 2B). However, comparisons of IC50 of HA extract determined for different exposure times showed concentration- and time-dependent cytotoxicity for both cancer cells (Fig 2E). Moreover, similar IC50 values were found which is visible in the heatmap. Where the IC50 values are missing, it was not possible to calculate them. These results highlight the potential of HA extracts to induce growth inhibitory effects, making it a promising candidate for further exploration of its anticancer properties.

Furthermore, the modulating activity of sub-inhibitory concentrations of HA extract (0.25 mg DW/mL) in combination with fluorouracil (5-FU) and L-Name on A549 and MDA-MB-231 cells (Fig 4C and 4D) was explored after a 24-hour exposure using the in vitro MTT assay. Based on the obtained data, statistically significant modulation was observed for both 5-FU and L-Name at the tested exposure times. Therefore, it was important to further understand the cellular mechanisms underlying the modulation of the cytotoxic properties of L-Name and 5-FU.

## 3.3 Analysis of Akt, PI3K, mTOR, and Caspase 3 expression in A549 and MDA-MB-231 cells

To elucidate the effects of various treatments on key signaling pathways involved in cancer cell proliferation and apoptosis, we analyzed the expression levels of Akt, PI3K, mTOR, and Caspase-3 in A549 and MDA-MB-231 cell lines. In A549 cells, PI3K levels decreased across all treatment groups, with a significant reduction observed in the HA + L-NAME and 5-FU groups (Fig 3A and 3B). Similarly, Akt levels were reduced in these two treatment groups (p < 0.05) (Fig 3A and 3B). In MDA-MB-231 cells, a comparable pattern was noted for PI3K levels, with an even more pronounced reduction. However, for Akt, a different outcome emerged: a

**Table 1. Docking results of HA major bioactive constituents on PI3K and Akt.**

| Ligand | PI3K Binding_Affinity | AKT Binding_Affinity | Total |
|---|---|---|---|
| Pseudohypericin | -12.7 | -7.6 | -20.3 |
| Chrysoeriol-glucuronide | -8.7 | -9.7 | -18.4 |
| Trihydroxyxanthone | -8.6 | -8.6 | -17.2 |
| Biapigenin | -8 | -8.9 | -16.9 |
| Quercetin | -8.4 | -8.5 | -16.9 |
| Myricetin_glucoside | -8.5 | -8.3 | -16.8 |
| Myricetin | -8.7 | -8 | -16.7 |
| References | -9.6 | -7 | -16.6 |
| Quercetin-acetyl-glucoside | -8.5 | -8 | -16.5 |
| Gallocatechin | -8.3 | -8 | -16.3 |
| Catechin | -8.5 | -7.8 | -16.3 |
| Quercetin-pentoside_2D_3D | -8.5 | -7.8 | -16.3 |
| Dihydromyricetin | -8.1 | -8.1 | -16.2 |
| Myricetin-arabinoside | -8.5 | -7.4 | -15.9 |
| Caffeoylquinic_acid | -7.8 | -8.1 | -15.9 |
| Epicatechin | -7.8 | -8.1 | -15.9 |
| p-Coumaroylquinic_acid | -7.7 | -7.7 | -15.4 |
| Galloyl_glucose | -7.7 | -7.6 | -15.3 |
| Quercetin-hexoside_2D_3D | -7.5 | -7.7 | -15.2 |
| Oleuropein | -7.9 | -7.3 | -15.2 |
| Vanillic_acid_glucoside | -7.3 | -7.5 | -14.8 |
| Carnosic_acid | -7.5 | -7.2 | -14.7 |
| Salicylic_acid_glucoside | -7.2 | -7.4 | -14.6 |
| Hypercalin_A | -7.5 | -7.1 | -14.6 |
| Methoxy-carnosic_acid | -7.4 | -6.9 | -14.3 |
| Hyperevolutin_A | -6.1 | -7.5 | -13.6 |
| Procyanidin_dimer | -6.3 | -7.1 | -13.4 |
| Trihydroxy_octadecadienoic_acid | -6.6 | -6.5 | -13.1 |
| Chipericumin_C | -5.6 | -6.7 | -12.3 |
| Pinellic_acid | -6.2 | -6.1 | -12.3 |
| hyperforing | -4.4 | -7 | -11.4 |
| Hyperpapuanone | -6.1 | -5.2 | -11.3 |
| adhyperforin | -4.2 | -6.8 | -11 |
| Quinic_acid | -5.2 | -5.2 | -10.4 |
| skyrin_glucoside_3D | -6.9 | -3.5 | -10.4 |
| Gluconic_acid | -4.2 | -4.7 | -8.9 |

significant decrease was observed exclusively in the HA + L-NAME combined treatment group compared to the control (Fig 3C and 3D).

In A549 cells, mTOR levels were reduced in the group treated solely with the plant extract (HA). In contrast, in MDA-MB-231 cells, treatment with the plant extract alone increased mTOR levels, whereas a reduction was observed in the groups treated with 5-FU or the combined treatments.

Caspase-3 levels were significantly increased in all treatment groups for both cell lines. In A549 cells, the most notable elevations were observed in the 5-FU ($p < 0.01$) and HA + L-NAME ($p < 0.05$) groups. A similar observation was made in MDA-MB-231 cells,

**Table 2. ADME properties of HA extract major phytochemicals.**

| Name | MW | HBA | HBD | LogP | Ro5 Viol |
|------|-----|-----|-----|------|----------|
| Pseudohypericin | 520.44 | 9 | 7 | 3.5 | 2 |
| Chrysoeriol glucuronide | 476.39 | 12 | 6 | 0.45 | 2 |
| Trihydroxyxanthone | 244.2 | 5 | 3 | 1.67 | 0 |
| Biapigenin | 538.46 | 10 | 6 | 3.62 | 2 |
| Quercetin | 302.24 | 7 | 5 | 1.23 | 0 |
| Myricetin_glucoside | 480.38 | 13 | 9 | -0.96 | 2 |
| Myricetin | 318.24 | 8 | 6 | 0.79 | 1 |

Abbreviations **used in the table:** MW (Molecular Weight); HBA (Hydrogen Bond Acceptors); HBD (Hydrogen Bond Donors); Ro5 Viol (Number of Lipinski's Rule of Five Violations).

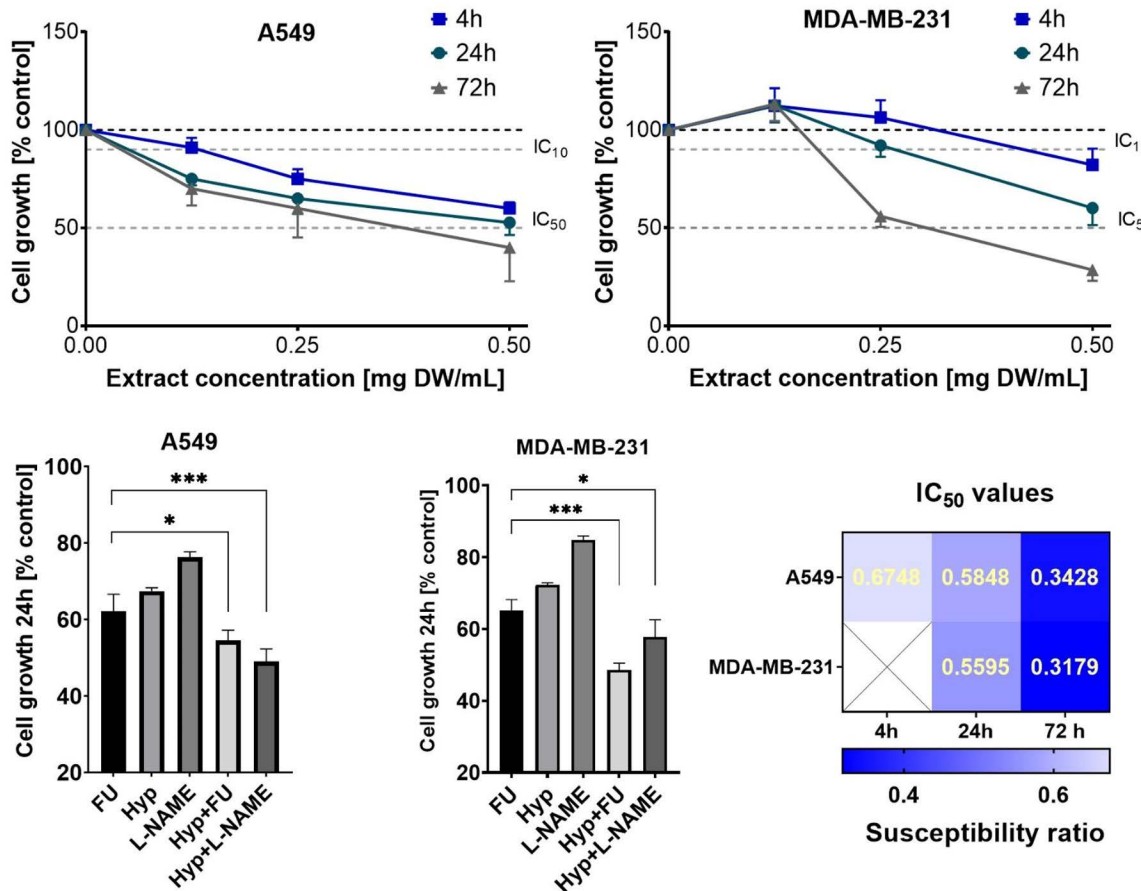

**Fig 2. Growth inhibition of A549 (panel A) and MDA-MB-231 (panel B) cells treated with HA extracts for 4, 24, and 72 hours, were assessed using the MTT assay.** The gray dashed lines in the graphs indicate the $IC_{10}$ and $IC_{50}$ levels, corresponding to 90% and 50% cell growth, respectively, compared to the control (100%). Panel E presents a heatmap of $IC_{50}$ values for the HA extract across different cell lines. The cytotoxicity-modulating effects of HA extract (0.25 mg DW/mL) in combination with L-NAME and/or 5-FU after a 24-hour exposure were evaluated in A549 (panel C), and MDA-MB-231 (panel D). Results are shown as means ± SD from three independent experiments, each performed in quadruplicate.

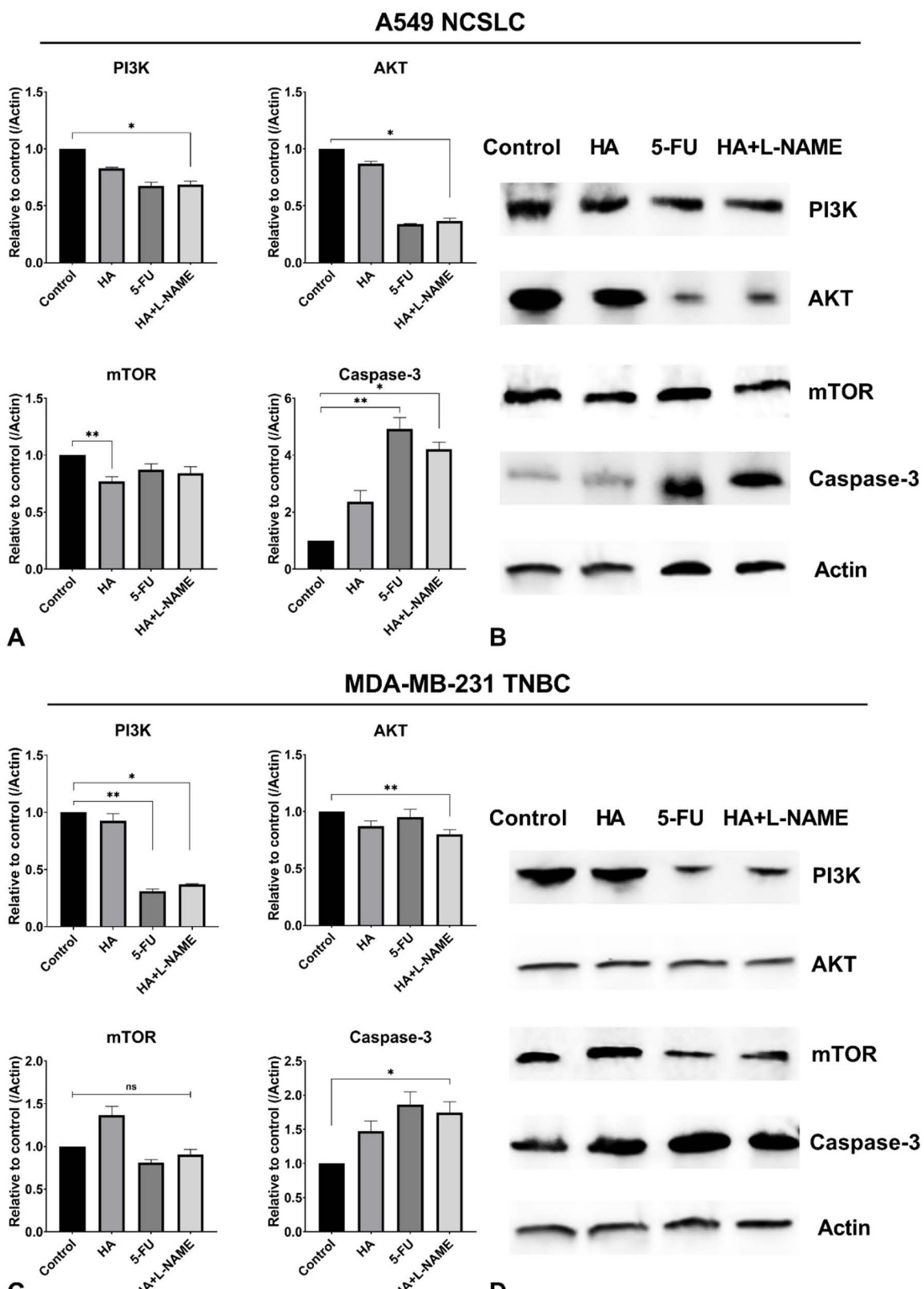

**Fig 3. Western blot analysis of Akt, PI3K, mTOR, and Caspase 3 expression in A549 (lung adenocarcinoma, NCSLC) and MDA-MB-231 (triple-negative breast cancer, TNBC) cell lines under different treatment conditions.** WB was performed to

assess the expression levels of Akt, PI3K, mTOR, and Caspase 3 in A549 and MDA-MB-231 cell lines, with β-actin used as a loading control. Panel A and B show the results for A549 cells, with Panel A displaying the quantification plots of each protein, and Panel B showing the corresponding Western blot bands. Panel C and D show the results for MDA-MB-231 cells, with Panel C displaying the quantification plots of each protein, and Panel D showing the Western blot bands. Cells were treated with *H. alpestre* plant extract, 5-FU (5-fluorouracil), and a combination of HA extract + L-NAME (a nitric oxide synthase inhibitor). The control group received no treatment.

with significant enhancements in the 5-FU and HA + L-NAME groups (p < 0.01) compared to the control (Fig 3). WB lines of expressed proteins were observed using the ImageJ program against Actin protein.

### 3.4  *H. alpestre* ethanolic extract alone and combined with L-NAME downregulates quantitative changes of TNFa, VEGFa, COX-2, and MMP-2

To clarify the factors influencing PI3K/Akt inhibition, we observed the quantitative changes in TNFα and VEGFα. *H. alpestre* extract reduced the quantity of TNFα in A549 cells by approximately two-fold (Fig 4A), and decreased the level of VEGFα by 2.7 times (Fig 4B). These two factors, interconnected with PI3K/Akt enzymes, influence and accept the decrease in their activity. To understand which compounds are affected by the signal after Akt, COX-2 and MMP-2 were considered. HA reduced the amount of COX-2 by about 30% (Fig 4C) and MMP-2 by 20% (Fig 4D). In the case of COX-2, the combination of *H. alpestre* and L-NAME is more potent (reducing its level by approximately five-fold) than their separate application (Fig 4C). The same pattern, but with less modulation of each other, is observed in the case of MMP-2. The amount of MMP-2 decreased by 35% under the influence of HALN (Fig 4D). In the next research phase, the potential effects of the herb extract on PI3K and Akt expression in A549 cancer cells were elucidated in vitro. The results demonstrated that both the herb extract alone and its combination with L-NAME significantly reduced the total amount of PI3K as well as its phosphorylated form (Fig 4E). This reduction indicates that the extract can influence both the enzymatic activity and gene expression of PI3K, thereby confirming the outcomes observed in the *in silico* study. Additionally, Akt analysis revealed that the herb extract decreased both the total and phosphorylated forms of the enzyme (Fig 4F).

### 3.5  Assessment of apoptosis

Hoechst 33258 stained apoptotic cells can be morphologically distinguished from viable cells by karyopyknosis, rough edges of their nuclei, and signs of nuclear fragmentation (Fig 5, **marked with arrows**).

All treatment variants induced a significant elevation of apoptosis rate in A549 cells compared to untreated control (Fig 5) (p < 0.05). In control cells, the apoptosis rate was 2.30 ± 1.51%, while 24 h incubation with 5-FU increased the rate of apoptotic cells to 15.61 ± 1.94% (Fig 5B and 5G). Treatments with HA or LN alone increased the rates of apoptotic cells up to 17.00 ± 1.41% and 17.20 ± 0.80%, respectively (Fig 5C, 5D and 5G). Treatment of cells with combinations of 5-FU+HA or HA+LN increased the rates of apoptotic cells up to 18.90 ± 0.70% and 27.7 ± 0.70%, respectively (Fig 5E and 5F). Thus, the apoptosis of A549 cells induced by the combination of HA+LN was significantly higher than that induced by 5-FU, HA, LN, and the combination of 5-FU+HA. (p < 0.05) (Fig 5G). We further evaluated the expression of active (cleaved) Caspase-3 using a colorimetric assay (Fig 5H). The increase in Caspase-3 activity was observed in all treatment variants which was more pronounced in cells treated with HA+LN (p < 0.05). For p53, an intense increase is observed in the 5-FU and HALN groups (Fig 5I).

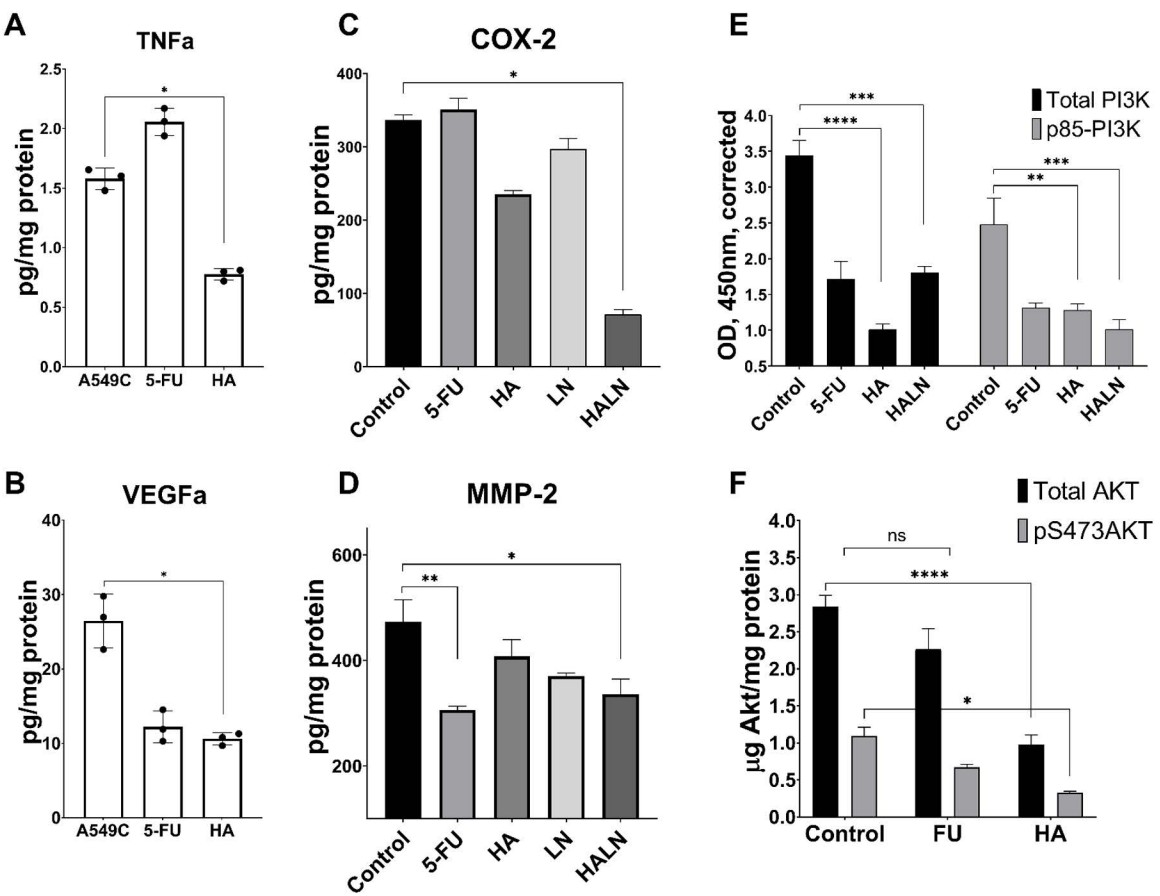

**Fig 4. The influence of *H. alpestre* extract alone and in combination with L-NAME on quantitative changes of TNFa (A), VEGFa (B), COX-2 (C), and MMP-2 (D) in A549 cells.** Control - A549C, 5-Fluorouracil - 5-FU (40 μM), *H. alpestre* - HA (0.25mg/mL), L-NAME – 14mM LN, HALN - HA + L-NAME (0.25mg/mL + 14mM). Three biologically independent repetitions of the experiments were performed, with two technical replicates each (n = 3, * - p ≤ 0.05, ** - p ≤ 0.01). Effect of HA, 5-FU, L-NAME, HA + L-NAME on PI3K/Akt pathway in A549 cells (E - PI3K, F - Akt). Total and phospho-kinases were each assayed in triplicate using the phospho- and total Kinase antibodies included in the PI 3 Kinase (E) and Akt (F) kits (n = 3, * - p ≤ 0.05, ** - p ≤ 0.01, *** - p ≤ 0.001, **** - p ≤ 0.0001). p85-PI3K - Phospho-PI 3 kinase p85, pS473AKT - phospho-Akt (Ser473).

We further validated the obtained results of cytotoxic properties of HA, LN, and their combination using MDA-MB-231 breast cancer cells. In control cells, the apoptosis rate was $5.80 \pm 0.85\%$, while 24 h incubation with 5-FU increased the rate of apoptotic cells to $13.20 \pm 1.13\%$ (Fig 6C and 6G). Treatments with HA or LN alone increased the rates of apoptotic cells up to $12.87 \pm 1.70\%$ and $40.30 \pm 2.69\%$, respectively (Fig 6B, 6D and 6G). Treatment of cells with combinations of 5-FU+HA or HA+LN increased the rates of apoptotic cells up to $18.74 \pm 1.29\%$ and $52.50 \pm 2.97\%$, respectively (Fig 6E and 6F). Thus, the apoptosis of MDA-MB-231 cells induced by the combination of HA+LN was significantly higher than that induced by 5-FU, HA, LN, and the combination of 5-FU+HA ($p < 0.05$) (Fig 6G).

## 4. Discussion

Our study demonstrates that ethanolic extract of *Hypericum alpestre*, particularly when combined with the nitric oxide synthase inhibitor L-NAME, significantly modulates the PI3K/Akt signaling pathway and its downstream effectors in human lung adenocarcinoma (A549) and

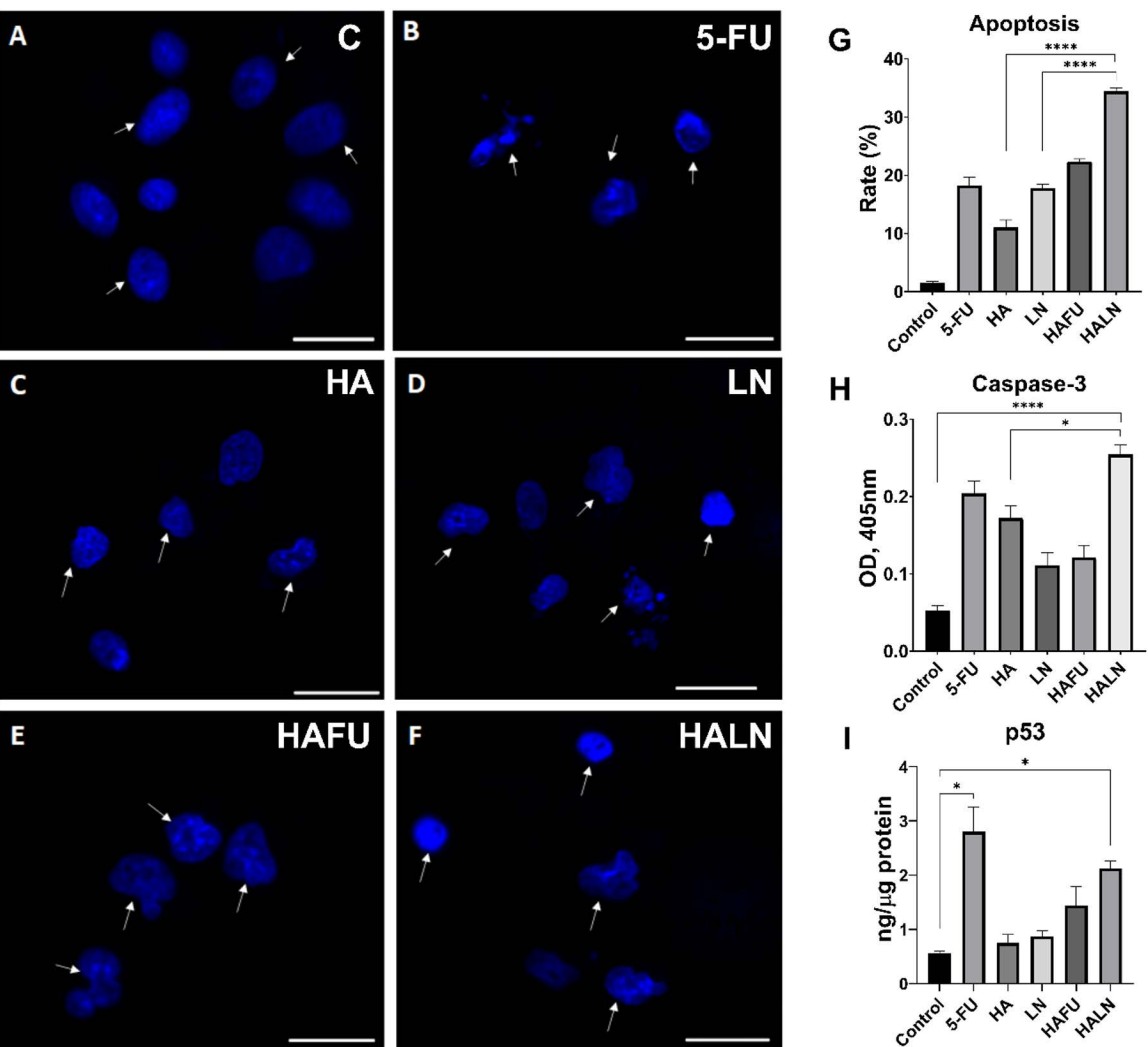

**Fig 5. Assessment of apoptosis by Hoechst 33258 staining (blue), caspase-3 activity (H), and p53 (I) quantity in A549 cells.**
Arrowheads indicate apoptotic cell nuclei. The scale bar is 100 μm. (A) Control cell nuclei have smooth edges. Treatment of cells with 5-FU (B), HA (C), LN (D), 5-FU+HA (E), and HA+LN (F) induced chromatin condensation, nuclear shrinkage, or fragmentation. (G) Frequencies of apoptotic cells were detected by morphological analysis using Hoechst 33258 staining, $^{*}p < 0.05$, $^{****}p < 0.0001$.

triple-negative breast cancer (MDA-MB-231) cells. *In silico* analyses, chrysoeriol glucuronide and pseudohypericin present in HA extract emerged as top candidates exhibiting strong affinities for PI3K and Akt. Further, we confirmed the interactions of HA extract with PI3K and Akt *in vitro*, showing that HA extract alone and in combination with L-NAME, reduced both total and phosphorylated forms of PI3K and Akt. These findings suggest that the bioactive polyphenols of HA can directly interface with critical oncogenic targets, thereby dampening a core survival and proliferation axis commonly implicated in various cancers [14,15].

A hallmark of the PI3K/Akt pathway is its involvement in orchestrating multiple oncogenic processes, including cell survival, metastasis, and angiogenesis [14,15,28]. In line with this, our data revealed a coordinated downregulation of pro-inflammatory (TNFα), pro-angiogenic (VEGFα), and metastasis-related (MMP-2, COX-2) factors following treatment with HA extracts. Notably, the combination of HA and L-NAME exhibited strong inhibition against

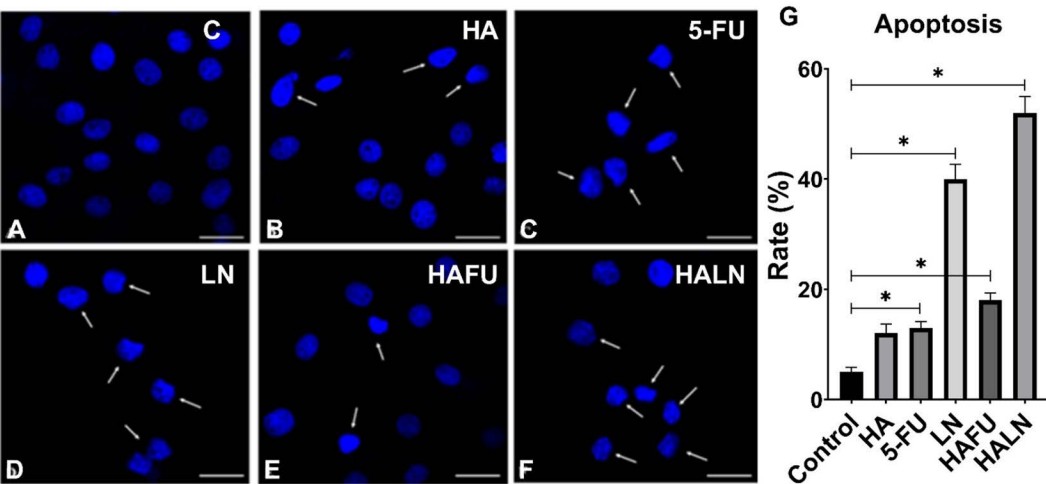

**Fig 6. Assessment of apoptosis by Hoechst 33258 staining (blue) in MDA-MB-231 cells.** Arrowheads indicate apoptotic cell nuclei. The scale bar is 100 μm. (A) Control cell nuclei have smooth edges. Treatment of cells with 5-FU (B), HA (C), LN (D), 5-FU+HA (E), and HA+LN (F) induced chromatin condensation, nuclear shrinkage, or blebbing. (G) Frequencies of apoptotic cells were detected by morphological analysis using Hoechst 33258 staining. * - p < 0.05.

COX-2 and MMP-2, suggesting that nitric oxide signaling may influence the extent to which HA-derived compounds exert their antitumor effects. By attenuating upstream mediators such as TNFα and VEGFα, and downstream effectors like COX-2 and MMP-2, HA extract and L-NAME co-treatment effectively disrupt multiple nodes of the oncogenic network, potentially limiting both the expansion and the invasive capacity of tumor cells.

This multi-level inhibition was further reflected in enhanced apoptotic activity. While HA alone induced apoptosis, the combination with L-NAME yielded significantly higher caspase-3 activation. This aligns with our hypothesis that modulating multiple aspects of tumor biology, key signaling pathways like PI3K/Akt, and associated inflammatory/angiogenic factors can create conditions more conducive to programmed cell death. Importantly, the HA + L-NAME combination outperformed the chemotherapeutic agent 5-FU in eliciting apoptosis, underscoring the potential of this natural extract–based regimen as a more selective and possibly less toxic alternative. Such findings resonate with emerging evidence that combining natural compounds with targeted inhibitors may overcome drug resistance and reduce side effects commonly associated with standard chemotherapy [7].

Western blot (WB) analysis revealed that HA inhibits mTOR in A549 cells, while its effects on PI3K and Akt in the same cells are minimal. These findings are further supported by ELISA results, which show a more pronounced lack of impact on PI3K and Akt. Additionally, HA enhances Caspase-3 activity, as demonstrated by both ELISA and WB analyses. In MDA-MB-231 cells, HA exhibits an activating effect on Caspase-3. However, among these observations, only the inhibition of mTOR in A549 cells is considered reliable. This may be due to compensatory mechanisms from other signaling pathways, such as the RAS/RAF pathway, which could offset the PI3K/Akt pathway. Nonetheless, the PI3K/Akt axis does not operate in isolation. Tumor cells often employ redundant or compensatory pathways, such as the RAS/RAF/MEK/ERK signaling cascade, to maintain survival even when PI3K/Akt is inhibited [29,30]. Although our current data support the potent inhibition of PI3K/Akt and its immediate downstream effectors, future studies should examine whether HA extract, alone or in combination with L-NAME, can counteract such compensatory

mechanisms. Elucidating these dynamics will be critical for optimizing therapeutic strategies and ensuring durable responses. Moreover, testing these regimens in animal models and clinical settings is warranted to evaluate pharmacokinetics, bioavailability, and potential synergistic benefits when combined with existing standard-of-care treatments. Another plausible explanation is the prooxidant property of this plant, as demonstrated in our previous study [6]. Given the plant's cytotoxic effects, shown in Fig 2, this property likely underlies its observed actions. The combination of the herb with L-NAME significantly inhibits PI3K and Akt in both cell lines (Fig 3A and 3C). Since mTOR occupies a central role in various signaling pathways, PI3K/Akt inhibition alone may not be sufficient to fully suppress its activity. Notably, the combination strongly enhances Caspase-3 activity, highlighting its potent apoptosis-inducing effects, which are also clearly evident in chromatin staining for both cell lines (Figs 5 and 6).

The relationship between NO and cancer has always been twofold [31,32]. Depending on the concentration and exposure time, it can have an apoptosis-promoting or cancer-promoting effect. In the case of cancer and other diseases, in particular, diabetes mellitus, hyperglycemia, or cardiovascular disorders the activity of NOS, changes in the amount of NO, and the interaction with the arginase enzyme are twofold and unclear [33,34]. In our previous work, we used L-NAME and the HA + L-NAME combination in an *in vivo* experimental model of breast cancer [6,11]. Many data in the literature and our results indicate that an 8-week L-NAME injection reduces the amount of NO, preventing the development of pathological angiogenesis. The use of the medicinal plant reduces oxidative stress, and therefore the decrease in peroxynitrite, which interferes with the bioavailability of NO, allowing NO to perform a beneficial action. The latter is confirmed by the results obtained in hyperglycemia, and prediabetes. Inhibition of arginase by L-norvaline, as well as additional provision of L-arginine to animals, leads to an increase in the amount of NO and its bioavailability, which inhibits the complications characteristic of diabetes in the cardiovascular system and kidneys [35]. In our previous work, we have shown that during hypoxia, a regulation of the activity of ornithine cycle enzymes occurs, which ensures the uninterrupted amount of NO as an adaptation factor to hypoxia [36]. Different studies show NO synthesis is observed with long-term use of L-NAME [33], and one of the processes may be this. At present, it should be noted that our previous *in vivo* studies have still shown that the use of L-NAME leads to a decrease in the amount of NO and RNS, which is one of the mechanisms of the anticancer effect [6,13,37].

Our findings highlight *Hypericum alpestre* extract, especially in combination with L-NAME, as a promising candidate for targeting the PI3K/Akt pathway and its associated oncogenic mediators (Fig 7). The study provides valuable insights into the molecular mechanisms underlying the anticancer effects of HA extract and L-NAME combination therapy. These findings support further preclinical and clinical investigations to evaluate the efficacy and safety of this combination in lung adenocarcinoma and other cancer types. Additionally, identifying specific compounds from HA extract with high affinity for PI3K/Akt proteins may facilitate the development of targeted therapies with improved efficacy and reduced side effects.

## 5. Conclusion

The combination of *Hypericum alpestre* (HA) aerial part extract with L-NAME shows significant anticancer potential, particularly in A549 lung adenocarcinoma and MDA-MB-231 breast cancer triple-negative cells. The study elucidated these properties' molecular mechanisms by focusing on the PI3K/Akt signaling pathway and its downstream

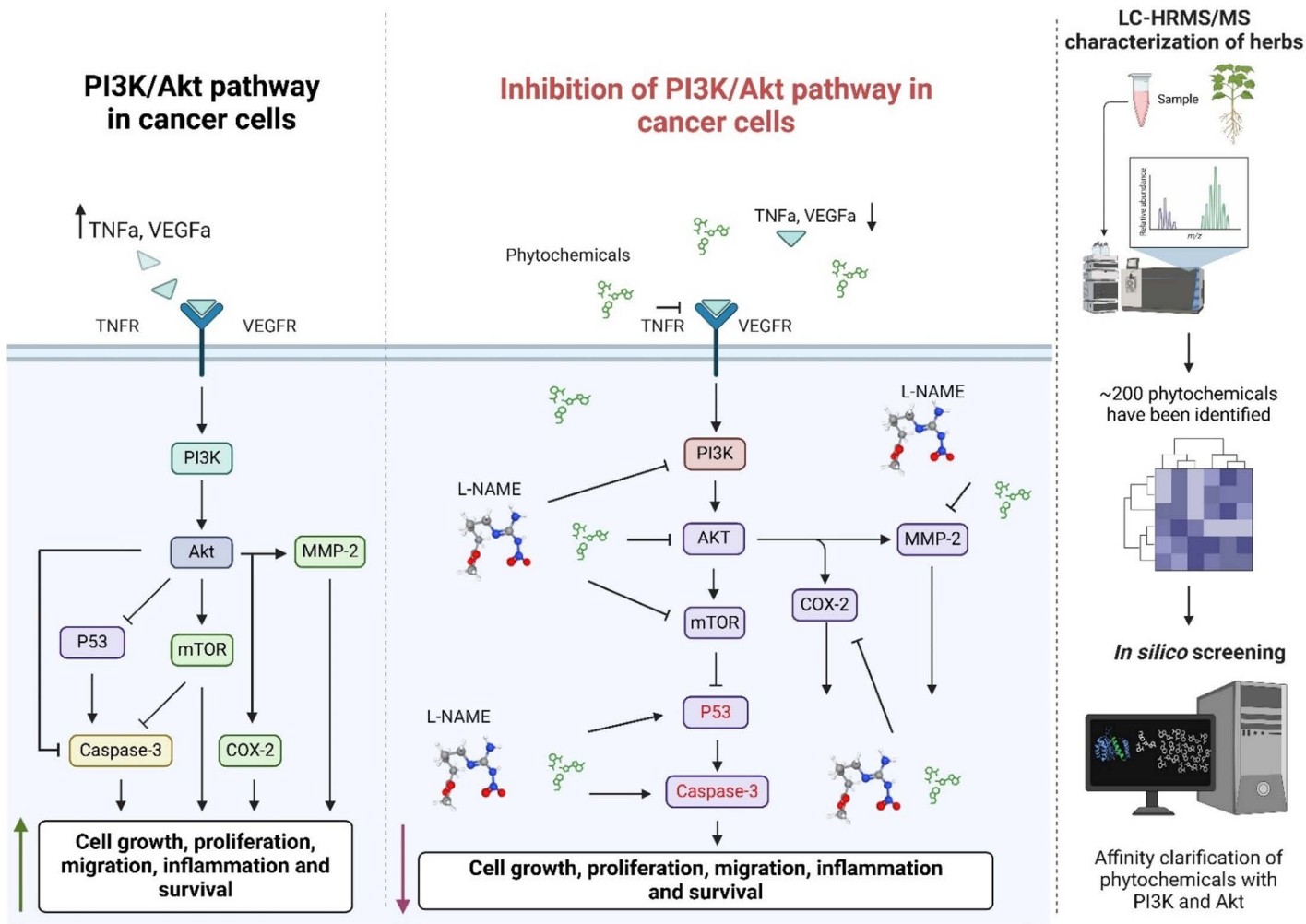

**Fig 7. Summary scheme of the obtained data.** *In silico* analysis of phytochemicals from the herbal plant *H. alpestre*, identified using LC-HRMS/MS, highlighted Chrysoeriol glucuronide and Pseudohypericin as the most active compounds. Experimental validation using WB, ELISA, and chromatin staining demonstrated that the combined treatment with the plant extract and the L-arginine metabolic pathway inhibitor L-NAME effectively inhibited the PI3K/Akt signaling pathway. This inhibition resulted in upregulated expression of P53 and Caspase-3. PI3K: Phosphoinositide 3-kinase; Akt: Protein Kinase B; TNFa: Tumor Necrosis Factor-alpha; VEGFa: Vascular Endothelial Growth Factor-alpha; TNFR: Tumor Necrosis Factor Receptor; VEGFR: Vascular Endothelial Growth Factor Receptor; L-NAME: Nω-Nitro-L-arginine methyl ester (an inhibitor of nitric oxide synthase); mTOR: Mammalian Target of Rapamycin; P53: Tumor Suppressor Protein 53; MMP-2: Matrix Metalloproteinase-2; COX-2: Cyclooxygenase-2; LC-HRMS/MS: Liquid Chromatography-High-Resolution Mass Spectrometry.

targets, including TNFa/COX-2 and VEGFa/MMP-2 pathways. *In silico* analysis identified specific compounds from HA extract, such as chrysoeriol glucuronide and pseudohypericin, with high affinity for PI3K and Akt proteins. Subsequent *in vitro* experiments confirmed that HA extract and its combination with L-NAME reduced the total amount and phosphorylated forms of PI3K and Akt, indicating their inhibitory effect on the PI3K/Akt pathway. Moreover, treatment with HA extract alone and in combination with L-NAME led to significant downregulation of TNFa, VEGFa, COX-2, and MMP-2, which are key components associated with cancer progression and angiogenesis. The observed increase in apoptosis, as evidenced by morphological changes and elevated Caspase-3 activity, further supports the efficacy of HA extract and L-NAME combination therapy inducing cancer cell death.

Overall, these findings suggest that the strong effects of HA extract and L-NAME on the PI3K/Akt signaling pathway and downstream targets hold promise for developing novel therapeutic strategies against cancer, particularly for drug-resistant tumors. Further research is warranted to validate these findings and explore the clinical potential of this combination therapy.

### 5.1 Limitations of the study

1. The use of a specific PI3K/Akt pathway inhibitor and the assessment of the PI3K regulator PTEN were not included.

2. The effects of the plant's pure phytochemical constituents were not studied.

3. The long-term effects of L-NAME remain unclear and require further investigation.

4. The RAS/RAF pathway, which is interconnected with the PI3K/Akt pathway, was not studied under the same experimental conditions.

## Supporting information

**S1 Table. Phytochemicals tentatively identified by LC-Q-Orbitrap-HRMS in *H. alpestre* aerial ethanol part extract.**
(DOCX)

**S1 File. All raw images of the western blot are in this manuscript.**
(RAR)

## Acknowledgments

Plant materials were identified by Dr. Narine Zakaryan from the Department of Botany and Mycology at Yerevan State University (YSU).

## Author contributions

**Conceptualization:** Hayarpi Javrushyan, Nikolay Avtandilyan.

**Data curation:** Alina Maloyan.

**Funding acquisition:** Tigran Harutyunyan, Nikolay Avtandilyan.

**Investigation:** Hayarpi Javrushyan, Mikayel Ginovyan, Tigran Harutyunyan, Smbat Gevorgyan, Zaruhi Karabekian.

**Methodology:** Hayarpi Javrushyan, Mikayel Ginovyan, Tigran Harutyunyan, Smbat Gevorgyan, Zaruhi Karabekian, Alina Maloyan.

**Project administration:** Nikolay Avtandilyan.

**Resources:** Mikayel Ginovyan, Nikolay Avtandilyan.

**Software:** Hayarpi Javrushyan, Tigran Harutyunyan, Smbat Gevorgyan.

**Supervision:** Alina Maloyan, Nikolay Avtandilyan.

**Validation:** Mikayel Ginovyan, Alina Maloyan.

**Visualization:** Tigran Harutyunyan, Smbat Gevorgyan, Zaruhi Karabekian.

**Writing – original draft:** Hayarpi Javrushyan, Mikayel Ginovyan, Tigran Harutyunyan, Smbat Gevorgyan, Zaruhi Karabekian, Alina Maloyan, Nikolay Avtandilyan.

**Writing – review & editing:** Mikayel Ginovyan, Alina Maloyan, Nikolay Avtandilyan.

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
