## [Decision Letter · Decision Letter 0]

11 Jun 2024

PONE-D-24-17168Elucidating the Impact of Hypericum alpestre Extract and L-NAME on the PI3K/Akt Signaling Pathway in A549 Lung Adenocarcinoma CellsPLOS ONE

Dear Dr. Avtandilyan,

Thank you for submitting your manuscript to PLOS ONE. After careful consideration, we feel that it has merit but does not fully meet PLOS ONE’s publication criteria as it currently stands. Therefore, we invite you to submit a revised version of the manuscript that addresses the points raised during the review process.

We look forward to receiving your revised manuscript.

Kind regards,

Vino Cheriyan, PhD

Academic Editor

PLOS ONE

Journal Requirements:

5. We note that your Data Availability Statement is currently as follows: All relevant data are within the manuscript and its Supporting Information files.

Additional Editor Comments:

The authors have effectively tackled the scientific query, offering ample background information and pertinent methodological details. The manuscript is well-crafted to enhance reader comprehension, however the authors should conduct essential and crucial experiments as suggested by the reviewers including the incorporation of additional lung cancer cell lines and performing confirmatory or supplementary western blots and other experiments as recommended.

Regarding the drug extract, kindly conduct a hemolytic test with the HA extract and please analyze the purity of the HA extract using HPLC.

In a study published in Pharmacol Rep 2012;64(3):511-20 by Jana Kopincova, that both short term and long term (acute and chronic) L-NAME treatment can affect blood pressure and vascular reactivity by reducing the availability of nitric oxide (NO). Interestingly, lower doses of L-NAME may trigger the production of NO through feedback mechanisms when used over an extended period. This phenomenon should be further discussed to provide a clearer understanding

Reviewers' comments:

Reviewer's Responses to Questions

**Comments to the Author**

1. Is the manuscript technically sound, and do the data support the conclusions?

Reviewer #1: Partly

Reviewer #2: Partly

Reviewer #3: Yes

2. Has the statistical analysis been performed appropriately and rigorously? 

Reviewer #1: Yes

Reviewer #2: Yes

Reviewer #3: Yes

3. Have the authors made all data underlying the findings in their manuscript fully available?

Reviewer #1: Yes

Reviewer #2: Yes

Reviewer #3: Yes

4. Is the manuscript presented in an intelligible fashion and written in standard English?

Reviewer #1: Yes

Reviewer #2: No

Reviewer #3: Yes

5. Review Comments to the Author

Reviewer #1: The authors are trying to uncover the anti-cancer effect of natural product HA extract, which is a promising strategy for cancer therapeutics targeting PI3/AKT signaling. How ever some critical experiments need to be addressed for successful publication

Please find my comments below

1. Authors needs to mention what dose the compounds are used and how they came up to the dose by showing viability assay with Ic50 values. If combination of HA and LA are cytotoxic to cells needs to be checked by a viability assay and colony formation ability as they are important indicators or cancer cell proliferation.

2. Authors can adopt wound healing assay with/with out treatment to show growth inhibition

3. I would recommend performing western blots as-well showing reduction in the PI3/AKT phosphorylation. I would like to know if the effects are temporary and if you do washout studies after treatment, will the cell recover? Also, a commercially available AKT /PI3 K inhibitor should be used as control.

4. At least 1 or two downstream signaling targets of PI3 /AKT is necessary to show ceasing of signaling pathway

5. Have you thought about looking the other arm of PI3 /AKT, which is the RAS/RAF pathway as a compensatory mechanism. Will the HA extract/combination affect RAS/RAF pathway effects, please mention your thoughts in the discussion

6. It would be nice to see schematic representation emphasizing where the compounds are targeting and downstream effects, would certainly enhance the publication quality

7. Please mention how Apoptosis rate was calculated. Visually Fig 3 B has more caspase activity (even though 3 cells), however 5-Fu has low apoptosis compared to other treatments, please elaborate the quantification

Reviewer #2: I recommend the inclusion of an additional cell line such as, Calu3 or NCI-H727 to validate the results obtained in A549 cells. The effect of HA, LN and combination of HA+LN was in A549 cells was reported, however, the docking studies have revealed the active ingredients within these extracts to be specific in the inhibitory effect on PI3K/AKT pathway. Is it possible to check the inhibitory effects of these active ingredients in these cells rather than the whole extracts? 5-FU is being used at a relatively higher concentration of 40uM in cells. Have the authors titrated the dose prior to its use in the assays? In Fig1 E and F, there is no data corresponding to LN, hence, the combined effect of HALN that is presented is left to speculation than a mere conclusion supported by evidence. Hence, effect of LN should also be included. How is the synergy observed upon treatment with HALN of Cox2, when the individual effects tend to be opposite? There is no observed synergy of HALN treatment on MMP2.

Overall, I recommend the inclusion of an additional cell line to validate the results and address the comments above.

Reviewer #3: The authors have done good job in addressing the scientific question. They have provided sufficient background inormation on the study, relevant details in the method section and detailed results significance. Manuscript is nicely written. For making the current study more comprehensible for the readers, authors are suggested to improve the quality of their figures, throughout the manuscript. This will enhance the clarity and overall significance of the study.

6. PLOS authors have the option to publish the peer review history of their article (what does this mean? ). If published, this will include your full peer review and any attached files.

**Do you want your identity to be public for this peer review?** For information about this choice, including consent withdrawal, please see our Privacy Policy .

Reviewer #1: **Yes: ** Sreeja C Sekhar

Reviewer #2: No

Reviewer #3: No

---

## [Author Response · Author response to Decision Letter 1]

18 Dec 2024

Nikolay Avtandilyan, PhD

Research Institute of Biology

Department of Biochemistry, Microbiology, and Biotechnology,

Yerevan State University, Yerevan, Armenia

Address: 1 Alex Manoogian, 0025, Yerevan, Republic of Armenia

Phone: (+374 10) 55-67-78,

Email: nv.avtandilyan@ysu.am

12.12.2024

EDITOR-IN-CHIEF

Emily Chenette

Dear Professor,

We would like to submit the revised original article “Elucidating the Impact of Hypericum alpestre Extract and L-NAME on the PI3K/Akt Signaling Pathway in A549 Lung Adenocarcinoma and MDA-MB-231 Triple-negative Breast Cancer Cells" for consideration for publication in PLOS ONE (PONE-D-24-17168). We very much appreciate the thoughtful review of our manuscript and have revised it to address all the reviewers’ suggestions and concerns. We believe that these revisions have greatly improved the quality of the paper. All revisions are “Track Changed”. We look forward to hearing from you.

Additional Editor Comments:

The authors have effectively tackled the scientific query, offering ample background information and pertinent methodological details. The manuscript is well-crafted to enhance reader comprehension, however the authors should conduct essential and crucial experiments as suggested by the reviewers including the incorporation of additional lung cancer cell lines and performing confirmatory or supplementary western blots and other experiments as recommended.

Comment: Regarding the drug extract, kindly conduct a hemolytic test with the HA extract and please analyze the purity of the HA extract using HPLC.

Answer: Thank you very much for those comments, which undoubtedly improved the quality of the manuscript. In our previous work, we performed UHPLC-ORBITRAP-HRMS analysis, which provided a detailed metabolic profile of the HA extract (Ginovyan et al., 2024). Additionally, during in vivo studies conducted earlier, we administered HA extract to healthy rats as a control group and observed no adverse effects indicative of hemolysis or other systemic toxicity (Ginovyan et al., 2024). Nevertheless, in line with the editor’s suggestions and to further ensure the safety and consistency of our preparations, we will conduct a hemolytic assay in upcoming experiments. UHPLC-ORBITRAP-HRMS analysis data are presented as a supplementary table.

Comment: In a study published in Pharmacol Rep 2012;64(3):511-20 by Jana Kopincova, that both short term and long term (acute and chronic) L-NAME treatment can affect blood pressure and vascular reactivity by reducing the availability of nitric oxide (NO). Interestingly, lower doses of L-NAME may trigger the production of NO through feedback mechanisms when used over an extended period. This phenomenon should be further discussed to provide a clearer understanding.

Answer: Thank you for a very interesting and correct observation. The relationship between NO and cancer has always been twofold. Depending on the concentration and exposure time, it can have an apoptosis-promoting or cancer-promoting effect. In the case of cancer and other diseases, in particular diabetes mellitus or hyperglycemia, the activity of NOS, changes in the amount of NO, and the interaction with the arginase enzyme are twofold and unclear. In our previous work, we used L-NAME and the HA+L-NAME combination in an in vivo experimental model of breast cancer. Many data in the literature and our results indicate that an 8-week L-NAME injection reduces the amount of NO, preventing the development of pathological angiogenesis. The use of the medicinal plant reduces oxidative stress, and therefore the decrease in peroxynitrite, which interferes with the bioavailability of NO, allowing NO to perform a beneficial action. The latter is confirmed by the results obtained in hyperglycemia, and prediabetes. Inhibition of arginase by L-norvaline, as well as additional provision of L-arginine to animals, leads to an increase in the amount of NO and its bioavailability, which inhibits the complications characteristic of diabetes in the cardiovascular system and kidneys. In our previous work, we have shown that during hypoxia, a regulation of the activity of ornithine cycle enzymes occurs, which ensures the uninterrupted amount of NO as an adaptation factor to hypoxia. Your observation that NO synthesis is observed with long-term use of L-NAME, one of the processes may be this. At present, it should be noted that our previous in vivo studies have still shown that the use of L-NAME leads to a decrease in the amount of NO and RNS, which is one of the mechanisms of the anticancer effect. Your observation will be the focus of our next in vivo studies, as, in essence, short-term and long-term effects can and should be observed in the animal experimental model itself. This section has been included in the Conclusion section, with all its references.

Review Comments to the Author

Reviewer #1: The authors are trying to uncover the anti-cancer effect of natural product HA extract, which is a promising strategy for cancer therapeutics targeting PI3/AKT signaling. How ever some critical experiments need to be addressed for successful publication

Please find my comments below

1. Authors needs to mention what dose the compounds are used and how they came up to the dose by showing viability assay with Ic50 values. If combination of HA and LA are cytotoxic to cells needs to be checked by a viability assay and colony formation ability as they are important indicators or cancer cell proliferation.

Answer: We agree that detailing the dose selection is critical. We have added MTT cytotoxicity results in the revised manuscript on both A549 and MDA-MB-231 cells which were used to determine the effective dose in further experiments (Figure 2). The IC50 values for the HA extract were established through MTT assays and mentioned in the revised manuscript in a separate section (3.2.Growth inhibiting properties of HA extract tested by MTT assay).

2. Authors can adopt wound healing assay with/with out treatment to show growth inhibition

Answer: We appreciate the reviewer’s suggestion. We have already purchased the required equipment and completed training for an auto-scratching wound healing assay. While these data are not included in the current manuscript, we will incorporate such assays in future studies to provide quantitative and reproducible evidence of HA extract’s anti-migratory effects.

3. I would recommend performing western blots as-well showing reduction in the PI3/AKT phosphorylation. I would like to know if the effects are temporary and if you do washout studies after treatment, will the cell recover? Also, a commercially available AKT /PI3 K inhibitor should be used as control.

Answer: We appreciate your suggestion to perform Western blots showing the reduction in PI3K/AKT phosphorylation to better assess the signaling pathway inhibition. In our current study, we focused on total PI3K and total AKT protein levels to analyze the general expression patterns (Figure 3). However, we recognize that phosphorylation is a key indicator of pathway activation and inhibition, and we plan to address this in future experiments by assessing phosphorylated forms of PI3K and AKT as you recommended. In this research work, we used the classic anticancer compound 5-FU, and we agree that the use of a specific inhibitor is desirable. We have acquired dactolisib (BEZ235) and will use it as a classic inhibitor in our in vivo studies. Thank you for the suggestion and for giving us a new direction. Additionally, this has been added as a limitation of the study.

4. At least 1 or two downstream signaling targets of PI3 /AKT is necessary to show ceasing of signaling pathway

Answer: Thank you for your helpful comment regarding the need for downstream signaling targets to demonstrate the cessation of the PI3K/AKT signaling pathway. To address the comment, we performed Western blot analysis for caspase-3 and mTOR in two different cell lines, namely triple-negative MDA-MB-231 breast cancer cells and A-549 lung adenocarcinoma cells. We analyzed the control group alongside three treatment groups: Hypericum alpestre plant extract, L-NAME + Hypericum alpestre combination, and 5-FU. These results have been included in the revised version of the manuscript to strengthen our conclusions about PI3K/Akt pathway inhibition (Figure 3).

5. Have you thought about looking the other arm of PI3 /AKT, which is the RAS/RAF pathway as a compensatory mechanism. Will the HA extract/combination affect RAS/RAF pathway effects, please mention your thoughts in the discussion.

Answer: Thank you for your insightful comment regarding the potential compensatory role of the RAS/RAF pathway in PI3K/AKT signaling. Yes, we are planning to focus on this metabolic pathway in our future research projects. Investigating how the HA extract/combination may impact the RAS/RAF pathway alongside PI3K/AKT signaling is an important next step. In response to your query, there are indeed studies indicating that PI3K/AKT and the RAS/RAF pathways are interconnected. For example, research has shown that inhibition of the PI3K/AKT pathway often leads to compensatory activation of the RAS/RAF pathway, suggesting a network of cross-talk between these two pathways in regulating cell growth and survival. One study by Morrison et al. (2006) in Nature Cell Biology demonstrates that PI3K/AKT inhibition can indeed trigger RAS/RAF/MEK/ERK signaling as a compensatory mechanism in cancer cells. This type of cross-regulation may be a significant factor when evaluating therapeutic interventions targeting PI3K/AKT. To rigorously address our hypotheses, we intend to undertake an in-depth investigation of the downstream signaling targets associated or interconnected with the PI3K/AKT pathway. We incorporated these considerations into the Discussion section and explored the possible interplay between these pathways in the context of HA extract/combination treatment.

6. It would be nice to see schematic representation emphasizing where the compounds are targeting and downstream effects, would certainly enhance the publication quality

Answer: Thank you for this useful observation. A diagram has been prepared, Figure 7, which presents the summary points of our study, the stages carried out, and the effects of the compounds on the PI3K/Akt pathway members under investigation.

7. Please mention how Apoptosis rate was calculated. Visually Fig 3 B has more caspase activity (even though 3 cells), however 5-Fu has low apoptosis compared to other treatments, please elaborate the quantification

Answer: The nuclei of cells were stained with Hoechst 33258 (which binds to chromatin) and morphologically analyzed by fluorescence microscope. Cells with smooth edges of nuclei were considered as non-apoptotic (Fig 5A) while cells with nuclear deformities or condensed chromatin were considered as apoptotic cells (Fig 5B-5F). For each variant, 500 cells were scored and the percentage of apoptotic cells was calculated as follows: % apoptotic cells = (the number of apoptotic cells/500 cells)*100%. Fig 5B is the representative image of apoptotic cell nuclei showing nuclear fragmentation and chromatin condensation after 5-FU treatment, analyzed by Hoechst 33258 staining, which does not allow determination of caspase activity. The level of caspase activity was determined using ELISA (Fig 3H). Thus, the frequency of apoptotic cells was determined by Hoechst 33258 staining while the level of active caspase enzyme was determined by ELISA and WB.

Reviewer #2:

1. I recommend the inclusion of an additional cell line such as, Calu3 or NCI-H727 to validate the results obtained in A549 cells.

Answer: As an additional cancer cell line we used MDA-MB-231 triple-negative breast cancer cells to validate the obtained results. We have chosen this cell line taking into account our earlier studies as well, where we demonstrated anticancer properties of HA alone and in combination with L-Name in vivo rat breast cancer model (Ginovyan et al. 2024, Hypericum alpestre extract exhibits in vitro and in vivo anticancer properties by regulating the cellular antioxidant system and metabolic pathway of L‐arginine, https://doi.org/10.1002/cbf.3914). This also allowed us to evaluate the efficiency of HA and its combination with L-NAME on different cancer types.

2. The effect of HA, LN and combination of HA+LN was in A549 cells was reported, however, the docking studies have revealed the active ingredients within these extracts to be specific in the inhibitory effect on PI3K/AKT pathway.

Answer: Thank you for the correct observation, yes, the extract was used in the research work, because at this stage we tried to clarify the mechanism of action, based on our previous in vivo studies. We have previously shown the antitumor, tissue transformation arrest, mortality reduction, and L-arginine metabolic pathway regulation effects of this herb, but the mechanism by which it acts has not yet been clarified. Next, we will study the effect of pure ingredients. We have also included this fact in the limitations of the study section.

3. Is it possible to check the inhibitory effects of these active ingredients in these cells rather than the whole extracts?

Answer: While our current work focuses on the HA extract as a whole, we recognize the value of testing the individual active compounds (e.g., chrysoeriol glucuronide and pseudohypericin) identified by in silico docking. We are planning to obtain these compounds and test their inhibitory effects on the PI3K/Akt pathway in future experiments.

4. 5-FU is being used at a relatively higher concentration of 40uM in cells. Have the authors titrated the dose prior to its use in the assays?

Answer: We used 40 μM 5-FU based on preliminary MTT cytotoxicity assays and selected concentrations with measurable cytotoxic effects within the timeframe of the experiments. These results are included in our earlier articles (https://doi.org/10.1038/s41598-024-65816-5).

5. In Fig1 E and F, there is no data corresponding to LN, hence, the combined effect of HALN that is presented is left to speculation than a mere conclusion supported by evidence. Hence, effect of LN should also be included.

Answer: There are quite a few research works related to L-NAME, as well as the works carried out by us. The effect of L-NAME alone has shown its NOS inhibitory abilities in vivo and in vitro models. It is known from the literature that Akt plays a crucial role in the regulation of NOS activity through phosphorylation. Many questions remain unclear as to how NOS inhibition affects downstream members, such as MMP-2 and COX-2. L-NAME does not affect Akt, Akt affects NOS. In the case of TNFa, and VEGFa, there cannot be a direct effect of L-NAME, it can affect the factors responsible for their synthesis. For this reason, it was advisable to conduct separate observations of L-NAME on the components presented in Figure 4. Despite all this, we agree that to obtain a complete and correct picture, it is desirable to determine the individual effect of L-NAME on essentially all targets. The alteration of various NOS-mediated pathways by the effect is quite well studied, and we wanted to consider those that are less, or not at all, defined.

6. How is the synergy observed upon treatment with HALN of Cox2, when the individual effects tend to be opposite? There is no observed synergy of HALN treatment on MMP2.

Answer: We have changed that section in the manuscript as follows: «In the case of COX-2, the combination of H. alpestre and L-NAME is more potent (reducing its level by approximately five-fold) than their separate application (Fig. 4, C). The same pattern, but with less modulation of each other, is observed in the case of MMP-2. The amount of MMP-2 decreased by 35% under the influence of HALN (Fig. 4, D)”.

7. Overall, I recommend the inclusion of an additional cell line to validate the results and address the comments above.

Answer: The use of the MDA-MB-231 triple-negative breast cancer cell line allowed us to additionally confirm our results on other cancer types as well.

Reviewer #3: The authors have done good job in addressing the scientific question. They have provided sufficient background inormation on the

---

## [Decision Letter · Decision Letter 1]

16 Jan 2025

Elucidating the Impact of Hypericum alpestre Extract and L-NAME on the PI3K/Akt Signaling Pathway in A549 Lung Adenocarcinoma and MDA-MB-231 Triple-negative Breast Cancer Cells

PONE-D-24-17168R1

Dear Dr. Nikolay Avtandilyan,

We’re pleased to inform you that your manuscript has been judged scientifically suitable for publication and will be formally accepted for publication once it meets all outstanding technical requirements.

Kind regards,

Vino Cheriyan, PhD

Academic Editor

PLOS ONE

Additional Editor Comments (optional):

The authors have revised the manuscript satisfactorily, responded to the majority of the comments and manuscript has been modified wherever necessary, I suggest that this work be published.

Reviewers' comments:

Reviewer's Responses to Questions

**Comments to the Author**

1. If the authors have adequately addressed your comments raised in a previous round of review and you feel that this manuscript is now acceptable for publication, you may indicate that here to bypass the “Comments to the Author” section, enter your conflict of interest statement in the “Confidential to Editor” section, and submit your "Accept" recommendation.

Reviewer #1: All comments have been addressed

Reviewer #2: All comments have been addressed

Reviewer #3: All comments have been addressed

2. Is the manuscript technically sound, and do the data support the conclusions?

Reviewer #1: Yes

Reviewer #2: Yes

Reviewer #3: Yes

3. Has the statistical analysis been performed appropriately and rigorously? 

Reviewer #1: Yes

Reviewer #2: Yes

Reviewer #3: Yes

4. Have the authors made all data underlying the findings in their manuscript fully available?

Reviewer #1: Yes

Reviewer #2: Yes

Reviewer #3: Yes

5. Is the manuscript presented in an intelligible fashion and written in standard English?

Reviewer #1: Yes

Reviewer #2: Yes

Reviewer #3: Yes

6. Review Comments to the Author

Reviewer #1: Authors have addressed all concerns raised previously and I recommend this manuscript for publication.

Reviewer #2: The authors have addressed all of the comments raised in the initial review. The authors have clarified the questions and revised the manuscript at places as required.

Reviewer #3: Manuscript is revised satisfactorally. Most of the comments have been addressed by the authors. Recommended to accept.

7. PLOS authors have the option to publish the peer review history of their article (what does this mean? ). If published, this will include your full peer review and any attached files.

**Do you want your identity to be public for this peer review?** For information about this choice, including consent withdrawal, please see our Privacy Policy .

Reviewer #1: No

Reviewer #2: No

Reviewer #3: No

---

## [Editor Report · Acceptance letter]

PONE-D-24-17168R1

PLOS ONE

Dear Dr. Avtandilyan,

I'm pleased to inform you that your manuscript has been deemed suitable for publication in PLOS ONE. Congratulations! Your manuscript is now being handed over to our production team.

Kind regards,

on behalf of

Dr. Vino Cheriyan

Academic Editor

PLOS ONE